# Development and Optimization of Nasal Composition of a Neuroprotective Agent for Use in Neonatology after Prenatal Hypoxia

**DOI:** 10.3390/ph17080990

**Published:** 2024-07-26

**Authors:** Igor Belenichev, Olena Aliyeva, Bogdan Burlaka, Kristina Burlaka, Oleh Kuchkovskyi, Dmytro Savchenko, Valentyn Oksenych, Oleksandr Kamyshnyi

**Affiliations:** 1Department of Pharmacology and Medical Formulation with Course of Normal Physiology, Zaporizhzhia State Medical and Pharmaceutical University, 69000 Zaporizhzhia, Ukraine; i.belenichev1914@gmail.com (I.B.);; 2Department of Histology, Cytology and Embryology, Zaporizhzhia State Medical and Pharmaceutical University, 69000 Zaporizhzhia, Ukraine; 3Department of Medicines Technology, Zaporizhzhia State Medical and Pharmaceutical University, 69000 Zaporizhzhia, Ukraine; 4Department of Clinical Laboratory Diagnostics, Zaporizhzhia State Medical and Pharmaceutical University, 69000 Zaporizhzhia, Ukraine; 5Department of Pharmacy and Industrial Drug Technology, Bogomolets National Medical University, 01601 Kyiv, Ukraine; 6Broegelmann Research Laboratory, Department of Clinical Science, University of Bergen, 5020 Bergen, Norway; 7Department of Microbiology, Virology and Immunology, I. Horbachevsky Ternopil State Medical University, 46001 Ternopil, Ukraine; kamyshnyi_om@tdmu.edu.ua

**Keywords:** prenatal hypoxia, neuroprotection, angiolin, intranasal gel

## Abstract

The intranasal route of drug administration is characterized by high bioavailability and is considered promising for rapid delivery of drugs with systemic action to the central nervous system (CNS), bypassing the blood-brain barrier. This is particularly important for the use of neuroprotective drugs in the treatment of brain tissue damage in infants caused by the effects of intrauterine hypoxia. The creation of new dosage forms for neonatology using mathematical technologies and special software in pharmaceutical development allows for the creation of cerebroprotective drugs with controlled pharmaco-technological properties, thus reducing time and resources for necessary research. We developed a new nasal gel formulation with Angiolin using a Box-Behnken experiment design for the therapy of prenatal CNS damage. It was found that the consistency characteristics of the nasal gel were significantly influenced by the gelling agent and mucoadhesive component—sodium salt of carboxymethylcellulose. We optimized the composition of nasal gel formulation with Angiolin using the formed models and relationships between the factors. The optimized nasal gel composition demonstrated satisfactory thixotropic properties. The 1% gel for neuroprotection with Angiolin, developed for intranasal administration, meets all safety requirements for this group of drug forms, showing low toxicity and no local irritant or allergic effects.

## 1. Introduction

The intranasal route of administration of active pharmaceutical ingredients is promising not only for the use of drugs for topical application but also for drugs of systemic action. This route of administration has become especially significant for drug delivery directly to the central nervous system (CNS), bypassing the blood-brain barrier, which was a traditionally difficult task for other dosage forms. This opens new possibilities for achieving positive therapeutic effects in the treatment of neurological disorders, especially neurodegenerative diseases, acute ischemic stroke, Alzheimer’s disease, and perinatal CNS damage. The intranasal route of administration is often equated with the parenteral route; it differs from other routes by high bioavailability, absence of first-pass metabolism of active substances, and potential use in neonatology and in unconscious patients [1,2,3,4].

The intranasal administration of active substances for the treatment of cerebrovascular diseases is based on the peculiarities of the anatomical structure of the nasal cavity. It is currently believed that absorption of active pharmaceutical ingredients occurs via the olfactory and trigeminal nerves. The olfactory pathway allows drugs to directly enter the olfactory bulb via the olfactory epithelium [5,6]. This pathway is critical for substance transport to both anterior and posterior brain regions due to entry points provided by these nerves. There is also a theory that drug transport is accomplished by both extracellular and intracellular pathways, with extracellular transport occurring by paracellular diffusion and intracellular transport by transcytosis [3,7,8,9,10].

Despite the promise of using the intranasal route to transport active pharmaceutical ingredients, several factors may hinder penetration: poor absorption through the nasal mucosa, possible enzymatic degradation, and removal of active substances. To overcome these obstacles, additional strategies have been proposed, such as the use of protease inhibitors, adsorption enhancers, mucoadhesive polymers, gel forms, and specific nano- and microemulsion technologies in nasal formulation [11,12,13].

The development of cerebral insufficiency in children who have experienced prenatal and perinatal hypoxia is rooted in neurodestructive processes linked to transmitter autocoidosis, oxidative and nitrosative stress, energy metabolism disorders, mitochondrial dysfunction, and inhibition of endogenous neuroprotective systems [14,15,16,17]. A promising approach to neuroprotection after prenatal hypoxia involves using agents that activate GSH/HSP70-dependent mechanisms of endogenous neuroprotection. In this context, a new molecule (S)-2,6-diaminohexanoic acid 3-methyl-1,2,4-triazolyl-5-trioacetate (Angiolin) is of interest, as it has demonstrated the ability to regulate HSP70 expression and influence the thiol-disulfide system under conditions of brain and myocardium ischemia and chronic alcoholization [14,18].

Angiolin showed neuroprotective activity when it was administered to animals undergoing prenatal hypoxia. The results of the study demonstrated that Angiolin reduced lethality and neurological disorders in animals after intrauterine hypoxia and increased the expression of HSP70, HIF-1a, and NO-system parameters in the brain [19,20,21]. Angiolin belongs to class VI of toxicity and has a high safety profile (no cumulation of substances, no immunotoxicity, mutagenicity, carcinogenicity, teratogenicity, embryotoxicity). Absence of toxic effect of the drug at 180-day administration to animals was shown. The first stage of clinical trials of Angiolin as an anti-ischemic agent with a significant effect on cerebral vascular endothelium, myocardium, and metabolism (solution for injection) has been successfully conducted with the permission of the State Expert Center of the Ministry of Health of Ukraine [22,23].

Therapeutic neuroprotective strategies for CNS damage in premature infants are not yet available for clinical use, necessitating the development of new dosage forms and administration routes for neonatology [24,25]. In this regard, nasal dosage forms with neuroprotectors are particularly promising. For the first time, we have obtained encouraging results on the therapeutic effect of Angiolin in experimental prenatal hypoxia [19,21]. The obtained results are justification for further studies and the development of new medicinal forms. In view of the above, the development and optimization of a new nasal form with Angiolin for neuroprotection after intrauterine hypoxia is an actual and perspective task. Therefore, we have developed a new nasal form of the neuroprotective agent Angiolin for intranasal administration to newborn infants with prenatal CNS damage. We believe that the successful development of this dosage form will reduce the incidence and severity of CNS pathology resulting from prenatal hypoxia, as well as minimizing complications, disability, mortality, and hospitalization duration.

The aim of this work was to develop and optimize the composition of a new nasal form with the neuroprotective agent Angiolin for use in neonatology after prenatal hypoxia.

## 2. Results

### 2.1. Formulation Design

The initial phase of pharmaceutical development using experimental design involves the careful planning and execution of experiments. Experiment design relies on the statistical analysis of experimental data, ranging from basic methods such as the t-test for comparing two groups to more complex methods such as analysis of variance (ANOVA). ANOVA is particularly useful for the examination of data in experiments with one or more factors, including situations in which there are interactions between the factors. Statistical evaluation is important for improving the quality of the final product through a comprehensive understanding of the variables that play a role in the final product [26,27].

Planning of the optimal formulation of nasal gel with Angiolin was performed using a Box-Behnken plan, with three levels of factors (Na CMC, Tween-80, and D-panthenol) and three responses (pH, Viscosity test, Angiolin Release) (Table 1).

### 2.2. Organoleptic Characteristics

According to the design of the experiment, we obtained samples of nasal gels, ranging from transparent to slightly yellowish color of different density and consistency, and odorless.

### 2.3. Hydrogen Index (pH)

Many factors contribute to the chemical stability of a formulation, one of which is the hydrogen value. In addition to the chemical stability of the formulation composition, pH can also significantly affect the nasal mucosa and potentially cause additional pathological processes.

The results of the statistical analysis of the effect of variable factors on the pH of the obtained nasal gels are presented in Table 2.

The obtained results of pH determination of nasal gel samples do not change significantly according to the ANOVA statistical analysis for the mean model.

### 2.4. Rheological Characteristics

Viscosity at a given shear rate is one of the key rheological characteristics. This parameter permits a comparative evaluation of the consistency properties of experimental gels and an assessment of the potential impact of formulation excipients on these properties. The results of the determination of the consistency characteristics of nasal gels are shown in Table 3.

The statistical analysis ANOVA for the reduced linear model demonstrates the reliable influence of factor A (Na CMC) on the viscosity values of the experimental nasal gel samples (F-value > *p*-value). The relationship between the viscosity values of gel preparation with factors (x) (Figure 1) considering the coefficients is presented in the equation:**Viscosity** = 545.888 + 499.59875 × A

The therapeutic effectiveness of a nasal local-acting drug depends on the active pharmaceutical ingredients included in the drug formulation as well as the rate of drug release in the nasal cavity (Table 4).

The results of the ANOVA statistical analysis for the quadratic model demonstrate a significant effect of factors A (Na CMC) and C (Tween-80) on the intensity of Angiolin release from nasal gel (F-value > *p*-value). The relationship between the percentage of Angiolin release intensity and research factors (Figure 2, Figure 3 and Figure 4) is shown in the equation:**Release** = 64.33 − 6 × A + 0 × B + 2.25 × C − 0.7499 × AB − 1.25 × AC − 0.25 × BC + 2.45 × A^2^ + 1.45 × B^2^ − 1.54 × C^2^(1)

Then, optimization of nasal gel formulation was carried out using a Box-Behnken design with number characteristics to predict the optimal characteristics of nasal gel formulation with Angiolin. The optimization procedure was set up in Design Expert 13.0 software on the following objectives: Release—goal maximize, Viscosity—goal maximize. The obtained results of prediction variants are summarized in Table 5.

For further studies, we selected formulation No. 1 (Table 6) that had the highest desirability—0.39, release—63.25, and viscosity—1045.480 with the following formulation (Table 6):

The technology for manufacturing the resulting optimized composition of Angiolin nasal gel is as follows: To make 100 g of gel, mixtures A and B are prepared and then mixed.

Mixture A: 47.64 mL of purified water is measured, and 1.5 g of Sodium CMC is taken. The mixture is then heated to 70–80 °C and left for 1 h to allow the Sodium CMC to swell. After the mixture has cooled to room temperature (25 °C), 1.0 g of D-panthenol is added while stirring with a magnetic stirrer at low speed.

Mixture B: The remaining 47.64 mL of water is measured out. 1.0 g of Angiolin and 1.2 g of Tween-80 are dissolved in the water with magnetic stirring until a homogeneous solution with a faint yellow tint, characteristic of Tween-80, is obtained. 0.02 g of benzalkonium hydrochloride is then added to the mixture under constant stirring.

Finally, mixture B is added to mixture A under constant stirring, resulting in a homogeneous gel solution with a yellowish color.

The developed composition of the nasal gel is stable when stored in cool conditions of 8–15 °C for 6 months (its organoleptic properties, pH, and Angiolin concentration do not change).

After preparation of the optimized formulation of the nasal gel with Angiolin, its thixotropic properties were studied (Figure 5).

### 2.5. Pharmaco-Toxicological Research

The obtained results of the thixotropy test show that the experimental nasal gel has satisfactory thixotropic properties. Its structure restores after the applied force; in particular, the restoration was 69% after 10 s, 76% after 30 s, and 79% after 60 s. This recovery pattern helps predict the stability of the dosage form both immediately after preparation and after use. As a result of pharmaco-toxicological tests, it was found that administration of nasal gel in an appropriate volume did not cause the death of animals during the whole observation period (Table 7). Additionally, the use of the gel did not cause any behavioral or visual abnormalities in the animals.

The obtained results demonstrate that nasal gel with Angiolin belongs to class VI of toxicity. The appearance and behavior of experimental animals had no visible pathological changes on the 1st, 7th, and 14th days after a single intranasal administration of Angiolin gel. In the study of a possible local irritant effect of Angiolin gel, mild conjunctival redness was observed in 1 of 10 animals immediately after drug administration. No conjunctival changes were observed in the remaining 9 animals (Table 8). On days 2 and 3 after application of the drug, no positive reaction from the eye conjunctiva was detected in all animals, indicating the absence of an irritative action of this dosage form.

Thus, nasal gel with Angiolin has no local irritant effect. As the results of the studies showed, daily application of 0.5 g of Angiolin gel (0.5 g of gel contains 5 mg of Angiolin) to the clipped area (4 × 4 cm) of the lateral surface of the body of experimental animals for 5 days and following single application of 0.3 g of this gel did not cause the development of anaphylactic shock (Table 9). There were no signs of anaphylactic shock at 6, 12, and 24 h after the application of 0.3 g of gel.

Thus, 1% gel with Angiolin administered over 5 days does not cause allergic reactions of anaphylactic type.

## 3. Discussion

The process of fetal and neonatal development is an important determinant of future health status and quality of life [28]. Prenatal hypoxia represents the primary etiological factor in perinatal cerebral pathology and neonatal mortality. In the context of neonatal mortality, hypoxia is the second most prevalent cause, surpassed only by prematurity [29]. The modern view of the pathogenesis of CNS damage after prenatal hypoxia determines the use of a promising direction, namely, the use of drugs with neuroprotective effect. The mechanisms of neurodegeneration during acute prenatal hypoxia include transmitter autocoidosis, glutamate-calcium cascade, impaired nitroxidergic system, and energy deficiency. Additionally, in the post-hypoxic period, significantly prolonged in time, the following processes are observed: oxidative stress, secondary mitochondrial dysfunction, neuroapoptosis, neuroinflammation, as well as deficiencies in neurotrophic and cytoprotective factors [30,31]. The modern arsenal of doctors includes specific drugs for the treatment of the consequences of hypoxic-ischemic lesions of the CNS [32,33]. However, high mortality and disability rates in children indicate that these pharmacotherapeutic agents are not fully effective. Therefore, the priority task of modern pharmacology is the development of new methods of pharmacological correction of neurological disorders caused by prenatal hypoxia. Evolution provides compensatory adaptive mechanisms of the placenta-fetus system that increase the resistance of the fetal CNS. These include the activation of endogenous neuroprotective processes in hypoxic brain damage, which involve 70 kDa heat shock proteins, hypoxia-induced factors, and oxidized intermediates of the thiol-disulfide system [14,18,19]. All this determines the perspectives of the HSP70/SH systems as a possible target link for pharmacological action. The long-term studies of the neuroprotective effect of potential drugs that modulate HSP70/SH system after prenatal hypoxia have revealed the promise of further study of Angiolin. For the first time, we demonstrated high neuroprotective activity of Angiolin after prenatal hypoxia, aimed at reducing lethality in the early postnatal period, improving neural tissue metabolism, reducing nitrosative stress, normalizing the expression of nNOS and iNOS, and increasing the expression of HSP70 and HIF-1a in the neonatal brain [19,21]. There is an idea of enhancing the neuroprotective effect of drugs by nasal administration in addition to the standard route. Intranasal drug delivery is emerging as a reliable method to bypass the blood-brain barrier and deliver a wide range of therapeutic agents, including growth factors and stem cells, directly to the brain [34,35,36]. The nasal delivery of mesenchymal stem cells to the mouse brain and their migration pathway through the lamina cribrosa into the brain were demonstrated for the first time [37]. Neuroprotective and regenerative effects were shown in experimental neonatal models by intranasal administration of neurotrophic drugs [25].

Creation of new dosage forms for neonatology necessitates the use of software and mathematical technologies in pharmaceutical development. This approach not only reduces the time and resources required by researchers but also enables the creation of cerebroprotective drugs with controlled pharmaco-technological properties. Delivery of active pharmaceutical ingredients using the nasal-brain pathway is an interesting concept that makes it possible to neutralize the disadvantages of the oral route of administration and to obtain a rapid therapeutic effect due to the anatomical features of the transport of active substances. Despite the obvious advantages of nasal drug delivery, additional factors must be taken into account when developing new nasal formulations: the pH of the formulation, mucoadhesive properties (viscosity of the dosage form) to control the time the drug stays in the nasal cavity, and increased permeability of active pharmaceutical ingredients through the blood-brain barrier.

The results of experimental studies of Angiolin nasal gel formulations indicate a significant effect of the mucoadhesive component of the sodium salt of carboxymethyl cellulose (F-value 91.91 > *p*-value 0.0002) and the adsorption enhancer Twine-80 (F-value 12.93 > *p*-value 0.0156) on the intensity of Angiolin release from nasal forms, while the plasticizer d-panthenol did not have a significant effect (F-value 1.814 × 10^−14^ < *p*-value 1.0000). The excipients used did not significantly affect the pH value of 6.41 ± 0.0025 (F-value 0.5667 < *p*-value 0.7871), which in turn will minimize irritation of the nasal cavity.

Our results align with our preliminary studies on the development of dosage forms with neuroprotective agents—Ademol, Noopept, and IL-1β antagonists. Experimental results of these dosage forms confirmed their high neuroprotective activity and good safety profile [38,39,40]. These results justify further preclinical study of 1% nasal gel with Angiolin as a neuroprotective agent after intrauterine hypoxia.

## 4. Materials and Methods

### 4.1. Materials

Angiolin ((S)-2,6-diaminohexanoic acid 3-methyl-1,2,4-triazolyl-5-thioacetate) (Scientific and Technological Complex “Institute of Monocrystals” NAS of Ukraine) was used as an active component in the formulation of nasal gel. Adjuvants: D-panthenol, carboxymethylcellulose sodium salt, tween-80, benzalkonium chloride, purified water. The active and adjuvant ingredients of pharmaceutical purity obtained from NPF “SINBIAS” LLC and “Istok-Plus” LLC were used in the experiments.

### 4.2. Design of the Experiment

For planning the design of the experiment, we used the response surface methodology and Box-Behnken plan, a set of statistical techniques that are used to create a model and conduct an analysis of the responses affected by the optimization factors. The design optimization was performed on three factors (x), three levels (low, medium, and high), and three responses (y) (Table 10).

### 4.3. Characteristics of the Obtained Gel Samples

#### 4.3.1. pH Test

We weighed 10.0 g of gel, placed it in a measuring cylinder, and added it to 100 mL of purified water. The obtained mixture was stirred with a magnetic stirrer. Then we measured the pH using a 15OM pH meter (Gomel Plant of Measuring Equipment JSC, Gomel, Belarus) with a glass electrode.

#### 4.3.2. Biopharmaceutical Study of the Kinetics of Substance Release through a Semipermeable Membrane

It was performed by the method of equilibrium dialysis through semipermeable membrane—Cuprophan cellophane film, Type 150 pm, 11 ± 0.5 μm thick in vertical diffusion cells of a nine-position station Franz Cells (PermeGear, Inc., Hellertown, PA, USA). Each cell contains a donor chamber in which the tested nasal gel sample was placed and an acceptor chamber of 25 mL filled with dialysis solution (purified water). In order to stir the dialysis solution uniformly in the acceptor chamber, a magnetic stirrer was added. The stirring speed of the dialysis solution was 350 rpm.

The temperature in the nasal cavity is heterogeneous and depends on factors such as ambient temperature, humidity, general health of the human body, as well as other factors, and ranges from 27 to 37 °C. For this study, we limited ourselves to a lower temperature value in order to assess the dynamics of release at the minimum temperature value.

The temperature regime in the experimental studies was 27 ± 0.2 °C, which was ensured by using a circulating water thermostat HAAKE SC100-S5P (Thermo Scientific, Portsmouth, NH, USA).

The percentage of Angiolin release was determined after 15 min by quantification of Angiolin using a spectrophotometric technique on a UV 2600 spectrophotometer (Shimadzu, Nishinokyo Kuwabara-cho, Nakagyo-ku, Japan) at 238 nm according to the modified method [23]. Before measurement, 5 mL of dialysate was taken, placed in a 25 mL volumetric flask, and purified water was added to the mark. The concentration of active substances in the solution was determined by a calibration graph. The results determined from the graph were increased five times with dilution and thus calculated the concentration of Angiolin in the dialysate.

The planned design of the experiment is aimed at studying the effect of the studied factors on the responses obtained in order to obtain an optimized formulation of the nasal gel with Angiolin. The duration of 15 min is the conditional average duration of the drug in the human nasal cavity, taking into account the normal functioning of mucociliary clearance.

#### 4.3.3. Rheological Studies

Rheological studies of the experimental gels were performed on a modular compact rheometer Anton Paar MCR 302 in the oscillation regime. The CP50-1 cone-plate system (diameter 50 mm, cone angle 1 degree) SN71317 was used as measuring devices, which, compared to cylindrical devices, requires significantly less gel sample and can perform the planned tests in an oscillatory regime. The temperature in the experiments was provided by an integrated thermostat. (Peltier temperature control for concentric cylinder systems, C-PTD 200).

#### 4.3.4. Viscosity Test of Sample Gels

We weighed 5 g of gel, placed it on a plate, and used the built-in RheoCompass software (1.33.615) to place a cone at a distance of 0.1 mm from the plate. The viscosity values (mPa × s) were measured at a shear rate (γ) of 50 s^−1^. The temperature regime of the experiment was performed at 25 °C.

#### 4.3.5. Thixotropy Test

We weighed out 5 g of gel, placed it on a plate, and using the built-in software RheoCompass (1.33.615), placed a cone at a distance of 0.1 mm from the plate. The study of the speed of structure recovery of the experimental sample of nasal gel was performed by the method of direct study with three intervals of oscillation—rotation—oscillation by DIN SPEC 91143-2:2012 (the three-interval thixotropy test, 3ITT). The experiment was carried out in three stages. 1. The initial measurement interval was carried out at a low shear rate (0.1 s^−1^), characterizing the resting behavior of the sample. 2. The middle measurement interval was performed at a shear rate of 100 s^−1^, characterizing the sample behavior at the moment of application. 3. Final interval—the measurement was carried out at a low shear rate of −0.1 s^−1^. This interval characterizes how quickly the sample restores its structure (in what time).

### 4.4. Pharmaco-Toxicological Methods of Research

The experiments were performed on 26 2-month-old white female Wistar rats weighing 95–110 g, obtained from the nursery of the Institute of Pharmacology and Toxicology of the Academy of Medical Sciences of Ukraine. A 14-day quarantine (acclimation) period was implemented for all animals. During this period, daily examinations were conducted to assess behavioral and general conditions, and the animals in the cages were observed twice a day to monitor morbidity and mortality. Before the start of the experiment, animals that met the criteria for inclusion in the experiment were selected and randomly assigned to groups. Cages with animals were placed in separate rooms. Light regime: 12 h—light, 12 h—darkness. Air temperature was maintained within 19–25 °C, relative air humidity—50–70%. Air temperature and humidity were registered every day. A ventilation regime was established, providing about 15 room volumes per hour. Animals were kept in the usual vivarium conditions. The animals were housed in 550 × 320 × 180 mm polycarbonate cages with 660 × 370 × 140 mm galvanized steel lids and glass drinkers. Five rats were kept in each cage. Each cage was labeled with the test number, species, sex, animal numbers, and dose. The cages were placed on racks according to the dose levels and cage numbers indicated on the labels. All rats were fed ad libitum standard food for laboratory animals supplied by “Phoenix” (Ukraine). Water from the municipal water supply (after reverse osmosis and UV sterilization) was given without limitation. Alder sawdust (Alnus glutinosa), pre-treated by autoclaving, was used as bedding. All manipulations were performed in compliance with the European convention for the protection of vertebrate animals used for experimental and other scientific purposes (Strasbourg, 1986), Directive 2010/63/EU of the European Parliament and of the Council on the protection of animals used for scientific purposes, and the general ethical principles of animal experiments [41,42]. The ZSMPhU Commission on Bioethics decided to adopt the experimental study protocols and outcomes (Protocol No. 2, of 18 May 2023).

#### 4.4.1. Acute Toxicity Definition

The acute toxicity study of 1% Angiolin gel was performed in 6 animals. The tested gels were administered intranasally using a dosing syringe in the maximum allowable volume for this route of administration—0.4 mL. Within 14 days, the animals were observed for death and changes in the cardiovascular system, respiratory system, CNS, and locomotor activity.

#### 4.4.2. Evaluation of Allergizing and Skin-Resorptive Activity by the Method of Skin Applications

On the lateral surface of the bodies of 10 animals, the hair was clipped in an area of 4 × 4 cm. 0.5 g of gel was applied to this skin area, after which the animals were placed in individual cages for 4 h to prevent drug licking. Gel application was performed by 20 repeated skin applications 5 times per week. Skin reactions were recorded daily using a skin test grading scale. The first test was performed after 10 applications (if an allergy was detected, further application of the gel was stopped). In case of a negative or doubtful result, the number of applications was necessarily increased to 20. Evaluation of the results of the allergic activity of the gel by the method of skin applications was carried out according to the appropriate scale (Table 11).

#### 4.4.3. Examination of Local Irritant Effect (Conjunctival Test)

The conjunctiva of both eyes of experimental animals was applied with a dosage pipette of 0.01 mL of gel. Rats of the control group were injected with distilled water into the conjunctival sac. The observation was carried out for 3 days.

The reaction was evaluated on a scale: 0 points—no changes in conjunctival mucosa; 1 point—slight conjunctival redness; 2 points—conjunctival redness and edema.

#### 4.4.4. Examination of Active Dermal Anaphylaxis

On the lateral surface of the bodies of 10 animals, the hair was removed in a 4 × 4 cm area. 0.5 g of gel was applied to this skin area, after which the animals were placed for 4 h in individual cages to prevent licking of the drug. Sensitization of animals was detected 5 days after the last application of the preparation. For this, 0.3–0.5 g of gel was applied once to the ear skin. The anaphylactic shock intensity was recorded after 6, 12, and 24 h in points according to Weigle index: ++++—shock with lethal outcome; +++—shock of severe degree (asphyxia, general convulsions, the animal loses the ability to keep on its feet, falls on the side, does not die); ++—medium shock (small convulsions, significant symptoms of bronchospasm); +—slight shock (some restlessness, rapid breathing, scratching of the face, spontaneous urination, defecation, hair ruffled); 0—shock has not developed, its symptoms are absent.

### 4.5. Statistical Methods

The statistical calculations used in the study were performed with the standard statistical package included with the licensed software program “STATISTICA^®^ for Windows 6.0” (StatSoftInc., Tulsa, OK, USA, № AXXR712D83323214FAN5), “SPSS 16.0”, and “Microsoft Office Excel 2010”.

## 5. Conclusions

In this study, a new nasal gel formulation with Angiolin was developed for the treatment of neonatal CNS damage after prenatal hypoxia using a Box-Behnken experimental design. It was found that the consistency properties of the nasal gel were significantly influenced by the gelling agent and mucoadhesive component—sodium salt of carboxymethylcellulose. The nasal gel formulation with Angiolin was optimized using the formed models and relationships between the factors. The optimized nasal gel demonstrated satisfactory thixotropic properties and met safety requirements, showing low toxicity and no local irritant or allergic effects. These results suggest that the nasal gel with Angiolin is a promising neuroprotective therapeutic dosage form for neonatology, with potential to reduce the incidence and severity of CNS pathology caused by prenatal hypoxia.

### Perspectives for Further Research

Further research on the new nasal gel is necessary to study its microbiological purity and stability during long-term storage, as well as to investigate the dynamics of Angiolin release in vitro at different temperatures of 27–37 °C; and under conditions of natural and artificial aging. We plan to perform a comprehensive evaluation of the neuroprotective effect of Angiolin nasal gel in modeling prenatal hypoxia on the concentration of neurodegradation markers (NR2, S100, matrix metalloproteinase-9, gallanin), indices of endogenous neuroprotection mechanisms (HSP70i/HSP70e, HIF-1a), morpho-functional indices of hippocampal CA1 neurons, markers of neuroinflammation and neuroapoptosis, indices of oxidative and nitrosative stresses, as well as on the indices of orienteering and exploratory activity of animals after prenatal hypoxia and the results of their learning and memory in the radial labyrinth. The neuroprotective effect of Angiolin nasal gel will be evaluated in comparison with drugs used in neonatology (Citicoline, Cerebrolysin).

## Figures and Tables

**Figure 1 pharmaceuticals-17-00990-f001:**
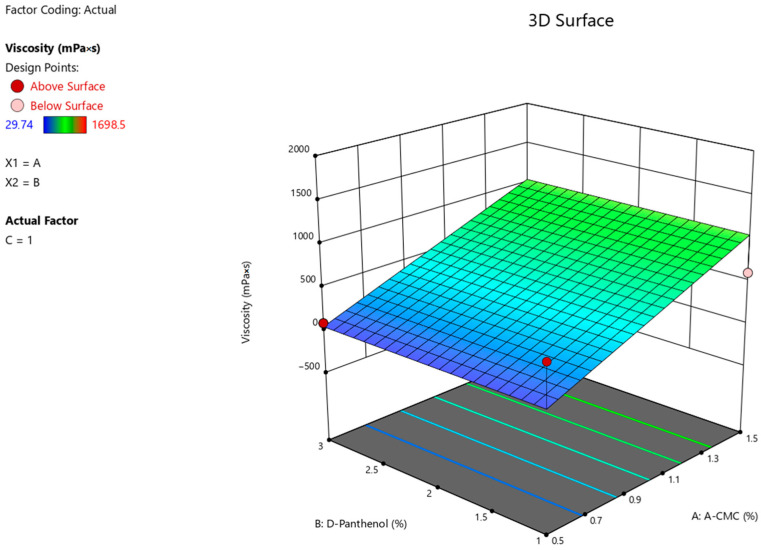
3D illustration of the relationship between the variable factors (Na CMC, D-panthenol) and viscosity characteristics of nasal gels.

**Figure 2 pharmaceuticals-17-00990-f002:**
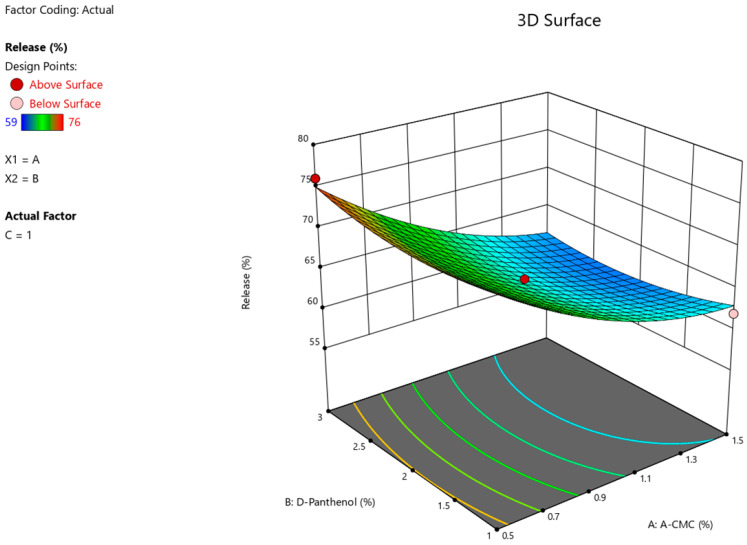
3D illustration of the relationship between the variable factors (Na CMC, D-Panthenol) and the percentage of Angiolin release intensity from nasal gels.

**Figure 3 pharmaceuticals-17-00990-f003:**
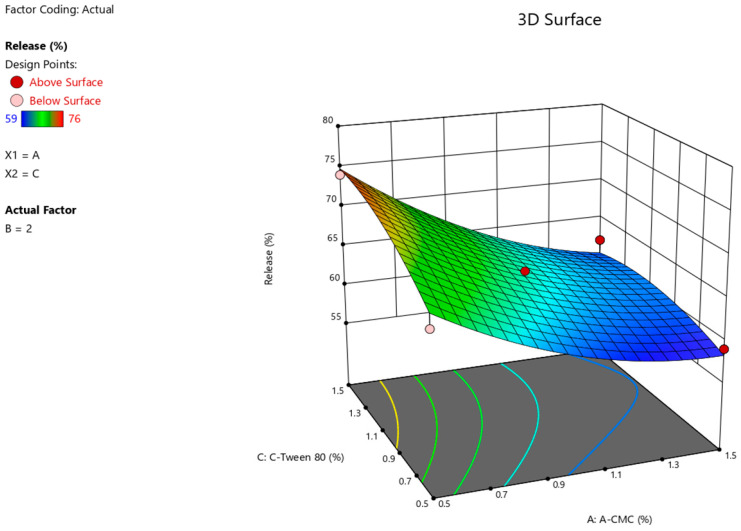
3D illustration of the relationship between the variable factors (Na CMC, Tween-80) and the percentage of Angiolin release intensity from nasal gels.

**Figure 4 pharmaceuticals-17-00990-f004:**
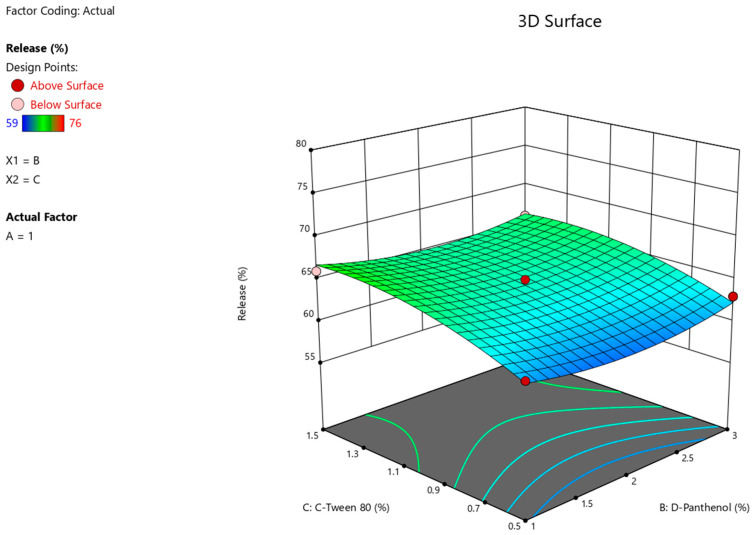
3D illustration of the relationship between modifying factors (Tween-80, D-Panthenol) and the percentage of Angiolin release intensity from nasal gels.

**Figure 5 pharmaceuticals-17-00990-f005:**
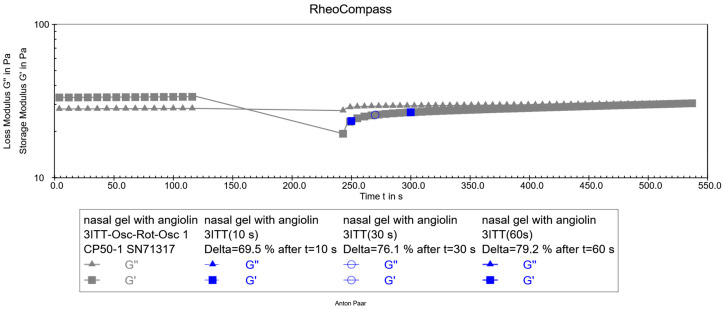
Thixotropy test of optimized nasal gel formulation with Angiolin.

**Table 1 pharmaceuticals-17-00990-t001:** Formulation design, factors, and responses received.

Run	Factor 1 A: CMC %	Factor 2 B: D-Panthenol, %	Factor 3 C: Tween 80, %	Response 1, pH	Response 2, Viscosity, mPa × s	Response 3 Release, %
1	1.5	3	1	6.41	782.15	60
2	0.5	2	1.5	6.41	80.01	74
3	1	2	1	6.41	432.57	65
4	1.5	2	0.5	6.41	1657.2	59
5	1	3	0.5	6.41	258.08	63
6	1	3	1.5	6.42	392.88	66
7	1	1	0.5	6.41	573.06	62
8	0.5	3	1	6.41	88.28	76
9	0.5	1	1	6.42	547.31	75
10	0.5	2	0.5	6.41	29.74	66
11	1.5	2	1.5	6.42	1698.5	62
12	1.5	1	1	6.41	604.28	62
13	1	1	1.5	6.41	443.38	66
14	1	2	1	6.41	161.23	64
15	1	2	1	6.42	439.65	64

**Table 2 pharmaceuticals-17-00990-t002:** Effect of variable factors on the pH value of nasal gels.

Source	Sum of Squares	df	Mean Square	F-Value	*p*-Value	Test Result
**Model**	0.0000	0				
**Residual**	0.0003	14	0.0000			
Lack of Fit	0.0002	12	0.0000	0.5667	0.7871	not significant
Pure Error	0.0001	2	0.0000			
**Cor Total**	0.0003	14				

**Table 3 pharmaceuticals-17-00990-t003:** Effect of variable factors (x) on the value of viscosity characteristics of nasal gels.

Source	Sum of Squares	df	Mean Square	F-Value	*p*-Value	Test Result
**Model**	1.997 × 10^6^	1	1.997 × 10^6^	16.14	0.0015	significant
A-A-CMC	1.997 × 10^6^	1	1.997 × 10^6^	16.14	0.0015	
**Residual**	1.608 × 10^6^	13	1.237 × 10^6^			
Lack of Fit	1.558 × 10^6^	11	1.416 × 10^6^	5.62	0.1606	not significant
Pure Error	50,397.74	2	25,198.87			
**Cor Total**	3.605 × 10^6^	14		16.14	0.0015	

**Table 4 pharmaceuticals-17-00990-t004:** Effect of variable factors (x) on the intensity of Angiolin release from nasal gels.

Source	Sum of Squares	df	Mean Square	F-Value	*p*-Value	Test Result
**Model**	377.93	9	41.99	13.40	0.0053	significant
A-A-CMC	288.00	1	288.00	91.91	0.0002	
B-D-Panthenol	5.684 × 10^−14^	1	5.684 × 10^−14^	1.814 × 10^−14^	1.0000	
C-C-Tween 80	40.50	1	40.50	12.93	0.0156	
AB	2.25	1	2.25	0.7181	0.4354	
AC	6.25	1	6.25	1.99	0.2170	
BC	0.2500	1	0.2500	0.0798	0.7889	
A^2^	22.31	1	22.31	7.12	0.0444	
B^2^	7.85	1	7.85	2.51	0.1743	
C^2^	8.78	1	8.78	2.80	0.1551	
**Residual**	15.67	5	3.13			
Lack of Fit	15.00	3	5.00	15.00	0.0631	not significant
Pure Error	0.6667	2	0.3333			
**Cor Total**	393.60	14				

**Table 5 pharmaceuticals-17-00990-t005:** Proposed formulation compositions of nasal gel with Angiolin.

Number	A-CMC, %	D-Panthenol, %	C-Tween 80, %	pH	Viscosity, mPa × s	Release, %	Desirability	Selected
1	1.500	1.000	1.203	6.413	1045.480	63.253	0.390	Selected
2	1.500	1.000	1.199	6.413	1045.483	63.252	0.390	
3	1.500	1.000	1.156	6.413	1045.484	63.240	0.390	
4	1.500	1.000	1.267	6.413	1045.485	63.228	0.389	
5	1.500	1.000	1.317	6.413	1045.485	63.173	0.387	
6	1.147	1.000	1.346	6.413	693.050	65.195	0.381	
7	1.147	1.000	1.347	6.413	693.083	65.195	0.381	
8	1.145	1.000	1.351	6.413	690.815	65.216	0.381	
9	1.149	1.000	1.352	6.413	695.111	65.176	0.381	
10	1.141	1.000	1.343	6.413	686.560	65.256	0.381	
11	1.158	1.000	1.337	6.413	703.316	65.101	0.381	
12	1.128	1.000	1.364	6.413	673.421	65.382	0.381	
13	1.111	1.000	1.356	6.413	656.933	65.547	0.380	
14	1.173	1.000	1.364	6.413	718.592	64.959	0.380	
15	1.097	1.000	1.361	6.413	642.599	65.694	0.380	
16	1.098	1.000	1.377	6.413	643.747	65.682	0.380	
17	1.221	1.000	1.318	6.413	766.358	64.569	0.380	
18	1.321	1.000	1.289	6.413	867.085	63.898	0.380	
19	1.238	1.000	1.295	6.413	783.634	64.437	0.380	
20	1.071	1.000	1.371	6.413	616.924	65.969	0.380	
21	0.982	3.000	1.331	6.413	527.538	66.722	0.368	
22	0.982	3.000	1.337	6.413	527.577	66.721	0.368	
23	0.978	3.000	1.334	6.413	523.881	66.779	0.368	
24	0.980	3.000	1.327	6.413	525.567	66.752	0.368	
25	0.983	3.000	1.327	6.413	528.520	66.706	0.368	
26	0.978	3.000	1.339	6.413	523.890	66.778	0.368	
27	0.982	3.000	1.315	6.413	528.122	66.711	0.368	
28	0.981	3.000	1.351	6.413	526.517	66.735	0.368	
29	0.994	3.000	1.355	6.413	539.887	66.527	0.368	
30	0.974	3.000	1.383	6.413	519.721	66.829	0.368	
31	0.959	3.000	1.286	6.413	505.317	67.054	0.367	
32	0.982	3.000	1.428	6.413	528.014	66.657	0.367	
33	1.027	3.000	1.260	6.413	572.949	66.021	0.367	
34	0.970	3.000	1.451	6.413	516.307	66.816	0.366	
35	0.970	3.000	1.109	6.413	516.073	66.581	0.361	
36	0.975	3.000	1.067	6.413	521.297	66.378	0.358	

**Table 6 pharmaceuticals-17-00990-t006:** Optimized formulation of nasal gel with Angiolin.

Name	Quantity (g)
Angiolin	1.0
Sodium CMC	1.5
D-panthenol	1
Benzalkonium chloride	0.02
Tween-80	1.2
Purified water	Up to 100

**Table 7 pharmaceuticals-17-00990-t007:** Study of acute toxicity of 1% nasal gel with Angiolin in rats by intranasal administration.

Volume, mL/100 g	Dose, mg/kg	Number of Rats			Lethality, %
		Total	Dead	Survived	
0.4	4	6	0	6	0

**Table 8 pharmaceuticals-17-00990-t008:** Results of the study of the local irritant effect of 1% gel with Angiolin.

Number of Animals	Period of the Study, Day					
	1		2		3	
	Control	Angiolin Gel	Control	Angiolin Gel	Control	Angiolin Gel
1	0	1	0	0	0	0
2	0	0	0	0	0	0
3	0	0	0	0	0	0
4	0	0	0	0	0	0
5	0	0	0	0	0	0
6	0	0	0	0	0	0
7	0	0	0	0	0	0
8	0	0	0	0	0	0
9	0	0	0	0	0	0
10	0	0	0	0	0	0

Notes: 0 points—no changes in conjunctiva; 1 point—slight reddening of conjunctiva.

**Table 9 pharmaceuticals-17-00990-t009:** Evaluation of active skin anaphylaxis of 1% gel with Angiolin in Weigle index score.

Number of Animals	Period of the Study, Hours					
	6		12		24	
	Control	Angiolin Gel	Control	Angiolin Gel	Control	Angiolin Gel
1	0	0	0	0	0	0
2	0	0	0	0	0	0
3	0	0	0	0	0	0
4	0	0	0	0	0	0
5	0	0	0	0	0	0
6	0	0	0	0	0	0
7	0	0	0	0	0	0
8	0	0	0	0	0	0
9	0	0	0	0	0	0
10	0	0	0	0	0	0

Notes: 0—shock has not developed; there are no signs of shock.

**Table 10 pharmaceuticals-17-00990-t010:** Factors and their levels used in the experiment design of the nasal gel with Angiolin.

Factor	Parameter	Levels		
		Low (−)	Medium (0)	High (+)
x1	Carboxymethylcellulose sodium salt, Na CMC, %	0.5	1	1.5
x2	D-panthenol %	1	2	3
x3	Tween-80, %	0.5	1	1.5

**Table 11 pharmaceuticals-17-00990-t011:** Scale for evaluation of application skin tests.

Reaction Designation	Symbols	Reaction Description
negative	-	no change in the skin
doubtful	±	small erythema without edema
weak-positive	+	erythema and edema at the application site
positive	++	erythema, edema, papules
strong positive	+++	erythema, edema, papules, isolated vesicles

## Data Availability

All the data generated during this research are included in the manuscript.

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
