# Peer review of "Development and Optimization of Nasal Composition of a Neuroprotective Agent for Use in Neonatology after Prenatal Hypoxia"

_pharmaceuticals, 2024, doi:10.3390/ph17080990_

Round 1

Reviewer 1 Report

Comments and Suggestions for Authors

Reviewer Comments:

 In this study author aim to develop and optimize the composition of a new nasal form with the neuroprotective agent Angiolin for use in neonatology after prenatal hypoxia. Results are interesting and this paper present the systematic approach to optimize the nasal gel as valuable canditiate for treating prenatal CNS damage. I would like to recommend this manuscript in the journal of Pharmaceuticals after these minor corrections.

 Comments:

1.     Author should highlight the novelty of this work in last paragraph of introduction.

2.     The Authors should add more relevant content in introduction as introduction is too short.

3.     Author should add comparative analysis of this nasal gel with other delivery methods or formulations for neuroprotection.

4.     Keeping in view the usage of neonatogy, author should investigate the durability of these gels.

5.     The authors are suggested to double-check the whole manuscript for grammar errors and typos that need to be corrected.

Comments on the Quality of English Language

The English language is ok. 

Author Response

We thank the Reviewer for the thorough and insightful evaluation of our article. We appreciate the positive feedback and constructive comments, which significantly improved the quality of the manuscript. We have made changes to the manuscript accordingly.

R1 Comments:

  1. Author should highlight the novelty of this work in last paragraph of introduction.

Answer1: Added.

  1. The Authors should add more relevant content in introduction as introduction is too short.

Answer2: The introduction was expanded.

  1. Author should add comparative analysis of this nasal gel with other delivery methods or formulations for neuroprotection.

Answer3: In the perspectives of further research: The neuroprotective effect of Angiolin nasal gel will be evaluated in comparison with drugs used in neonatology (Citicoline, Cerebrolysin).

  1. Keeping in view the usage of neonatogy, author should investigate the durability of these gels.

Answer4: In the perspectives of further research: Study of microbiological purity and stability during long-term storage, study of the dynamics of Angiolin release in vitro at different temperatures of 27-37 °C, and under conditions of natural and artificial aging.

  1. The authors are suggested to double-check the whole manuscript for grammar errors and typos that need to be corrected.

Answer5: We have now proofread the manuscript. Additional proofreading is professionally done by MDPI after the paper is accepted.

Reviewer 2 Report

Comments and Suggestions for Authors

Dear Editor,

I have reviewed the manuscript titled "Development and optimization of nasal composition of a neuroprotective agent for use in neonatology after prenatal hypoxia" While the study is well-executed and presents clear findings, I find the paper to be relatively simple and lacking in novelty. The focus on intranasal delivery for neuroprotective drugs is not a new concept, especially with a simple gel formulation. The authors should consider more advanced preparations, such as a nanogel, which could potentially offer better drug delivery efficiency and therapeutic outcomes. Additionally, the paper would benefit from including more comprehensive studies, such as histopathology, to provide a deeper understanding of the formulation's efficacy and safety. The method of preparation of the gel is also not mentioned, which is a critical aspect that needs to be detailed for reproducibility and validation of the study.

Here are specific comments that need to be addressed:

1.     Line 48, CNS, it first mentioned in the paper and should be defined.

2.     The method of preparation of the nasal gel is not mentioned in the manuscript. should be mentioned before the Characteristics of the obtained gel samples section

3.     Maintain the consistency of the capital letter (angiolin)( Angiolin) thorough the manuscript

4.     Line 161, sufficient number of animals, mention how many animals in each groups

5.     Line 180, Acute toxicity definition, mentions the dose of angiolin administered

6.     Line 187, 0.5 g of gel, how much equivalent to the drug

7.     Line 232, Table, column 6 response 1 ph is repeated ( viscosity )

8. In Table 3, the number of runs is 15, while in Table 6 the number of runs is 36 , why?

9.     Why not conducted stability and compatibility studies

10.  Discussion part is not enough.

11.  Rewrite the conclusion part

12.  Perspectives for further research, specify which studies are needed to investigate the specific activity, and what activity, for the drug or the formulation?

13.  Correct any minor grammatical issues and ensure consistency in terminology

Author Response

We thank the Reviewer for evaluating our manuscript

I have reviewed the manuscript titled "Development and optimization of nasal composition of a neuroprotective agent for use in neonatology after prenatal hypoxia" While the study is well-executed and presents clear findings, I find the paper to be relatively simple and lacking in novelty. The focus on intranasal delivery for neuroprotective drugs is not a new concept, especially with a simple gel formulation. The authors should consider more advanced preparations, such as a nanogel, which could potentially offer better drug delivery efficiency and therapeutic outcomes. Additionally, the paper would benefit from including more comprehensive studies, such as histopathology, to provide a deeper understanding of the formulation's efficacy and safety. The method of preparation of the gel is also not mentioned, which is a critical aspect that needs to be detailed for reproducibility and validation of the study.

Here are specific comments that need to be addressed:

  1. Line 48, CNS, it first mentioned in the paper and should be defined.

Answer1: Added.

  1. The method of preparation of the nasal gel is not mentioned in the manuscript. should be mentioned before the Characteristics of the obtained gel samples section

Answer2: The method of preparation of the nasal gel is now added (P. 8-9).

  1. Maintain the consistency of the capital letter (angiolin)( Angiolin) thorough the manuscript

Answer3: Corrected

  1. Line 161, sufficient number of animals, mention how many animals in each groups

Answer4: Added

  1. Line 180, Acute toxicity definition, mentions the dose of Angiolin administered

Answer5: The dose of 1% nasal gel with Angiolin for the acute toxicity definition is given in Table 7.

  1. Line 187, 0.5 g of gel, how much equivalent to the drug

Answer6: Added. 0.5 g of Angiolin gel (0.5 g of gel contains 5 mg of Angiolin).

  1. Line 232, Table, column 6 response 1 ph is repeated ( viscosity )

Answer7: Corrected

  1. In Table 3, the number of runs is 15, while in Table 6 the number of runs is 36 , why?

Answer8: The design of the experiment was done using design expert software. Table 3 characterizes the number of runs to determine the influence of the studied factors on the obtained responses, according to the programmed plan. In Table 6 the software offers possible variants of predicted compositions of nasal gel with Angiolin considering desirability, obtained statistical models, as well as selected optimization parameters.

  1. Why not conducted stability and compatibility studies?

Answer9: The developed nasal gel composition is stable when stored in cool conditions 8-15 ºC for 6 months (its organoleptic properties, pH and Angiolin concentration do not change). Further studies are being conducted to determine stability during long-term storage using natural and artificial aging techniques.

  1. Discussion part is not enough.

Answer 10: The discussion part has been expanded.

  1. Rewrite the conclusion part

Answer 11: The conclusions have now been revised

  1. Perspectives for further research, specify which studies are needed to investigate the specific activity, and what activity, for the drug or the formulation?

Response12: We have now included this information in the manuscript.

  1. Correct any minor grammatical issues and ensure consistency in terminology.

Answer13: We have now proofread the manuscript

We thank again the Reviewer for evaluating out manuscript

Reviewer 3 Report

Comments and Suggestions for Authors

The manuscript by Belenichev I et al. entitled: "Development and optimization of nasal composition of a neuroprotective agent for use in neonatology after prenatal hypoxia" contains an interesting study on the development and characterization of some angiolin-containing hydrogels for nasal application.

However, some questions need to be clarified:

-              Why did the author choose 27±0.2 °C as the temperature for release performance? It is recommended to perform the test at 37±0.2 °C.

-              Why was the percentage of angiolin release only determined after 15 minutes and the analysis was not continued until the maximum release was reached?

-              Column 6 of Table 3 is incorrect

-              The authors did not describe the preparation method for the hydrogels studied

-              In the pharmacotechnical evaluation, a lot of statistics are used, they are emphasized and the significance of the resulting values is minimized. They are not discussed but only presented in tables

-              The in vivo section completely lacks statistics and discussion

-              The results recorded for rheology and release properties need to be explained according to the influence of the ingredients

-              The discussion section needs to be expanded and the comparison of the results with similar studies from the literature needs to be made

Comments on the Quality of English Language

English language needs only minor revision

Author Response

We thank the Reviewer for evaluating our manuscript

The manuscript by Belenichev I et al. entitled: "Development and optimization of nasal composition of a neuroprotective agent for use in neonatology after prenatal hypoxia" contains an interesting study on the development and characterization of some Angiolin-containing hydrogels for nasal application.

However, some questions need to be clarified:

  • Why did the author choose 27±0.2 °C as the temperature for release performance? It is recommended to perform the test at 37±0.2 °C.
  • Response: 

    - The temperature in the nasal cavity is heterogeneous and depends on factors such as ambient temperature, humidity, general condition of the human body, and other factors, and ranges from 27 to 37 ºC. For this study, we chose the lower temperature value to evaluate the release dynamics at the minimum temperature value. In the future, we plan to conduct additional studies on the effect of temperature on the release rate of Angiolin.

  •  
  • Why was the percentage of Angiolin release only determined after 15 minutes and the analysis was not continued until the maximum release was reached?
  • Response: The planned design of our experiment is aimed at studying the influence of the studied factors on the obtained responses in order to obtain an optimized formulation of nasal gel with Angiolin. The duration of 15 minutes is a conditional average duration of the drug in the human nasal cavity, taking into account the normal functioning of mucociliary clearance. In the future we plan to show the dynamics of Angiolin release in vitro for a longer time taking into account different temperature regime 27-37 C.

  •  
  • Column 6 of Table 3 is incorrect
  • Response: Column 6 has now been corrected.

  •  
  • The authors did not describe the preparation method for the hydrogels studied
  • Response: The method of preparation of the nasal gel is now added (P. 8-9).

  •  
  • In the pharmacotechnical evaluation, a lot of statistics are used, they are emphasized and the significance of the resulting values is minimized. They are not discussed but only presented in tables
  • Response: These statistics characterize the obtained model, which was used to obtain the predicted nasal gel composition.

  •  
  • The in vivo section completely lacks statistics and discussion
  • Response:  Pharmacotoxicological studies were performed according to standard methods. Results were scored according to scales (described in Methods). These results do not require statistical analysis.

  • The results recorded for rheology and release properties need to be explained according to the influence of the ingredients
  • Response: Added to the Discussion part.

  • The discussion section needs to be expanded and the comparison of the results with similar studies from the literature needs to be made.
  • Response: Added to the Discussion part.

We thank again the review for evaluating our manuscript

Round 2

Reviewer 2 Report

Comments and Suggestions for Authors

The author responded to all my points and the paper is now improved and can be accepted 

Reviewer 3 Report

Comments and Suggestions for Authors

Thank you for the provided answers

Comments on the Quality of English Language

English Language needs only minor revision